

# Comparison of integrated PET/MRI with PET/CT in evaluation of endometrial cancer: a retrospective analysis of 81 cases

Li-hua Bian, Min Wang, Jing Gong, Hong-hong Liu, Nan Wang, Na Wen, Wen-sheng Fan, Bai-xuan Xu, Ming-yang Wang, Ming-xia Ye and Yuan-guang Meng

Department of Gynecology and Obstetrics, General Hospital of PLA, Beijing, China

## ABSTRACT

**Background.** The objective of this study was to compare the diagnostic value of integrated PET/MRI with PET/CT for assessment of regional lymph node metastasis and deep myometrial invasion detection of endometrial cancer.

**Methods.** Eighty-one patients with biopsy-proven endometrial cancer underwent preoperative PET/CT ($n = 37$) and integrated PET/MRI ($n = 44$) for initial staging. The diagnostic performance of PET/CT and integrated PET/MRI for assessing the extent of the primary tumor and metastasis to the regional lymph nodes was evaluated by two experienced readers. Histopathological and follow-up imaging results were used as the gold standard. McNemar's test was employed for statistical analysis.

**Results.** Integrated PET/MRI and PET/CT both detected 100% of the primary tumors. Integrated PET/MRI proved significantly more sensitivity and specificity than PET/CT in regional lymph node metastasis detection ($P = 0.015$ and $P < 0.001$, respectively). The overall accuracy of myometrial invasion detection for PET/CT and Integrated PET/MRI was 45.9% and 81.8%, respectively. Integrated PET/MRI proved significantly more accurate than PET/CT ($P < 0.001$).

**Conclusion.** Integrated PET/MRI, which complements the individual advantages of MRI and PET, is a valuable technique for the assessment of the lymph node metastasis and myometrial invasion in patients with endometrial cancer.

## INTRODUCTION

Endometrial cancer is the sixth most common cancer in women worldwide and the most common gynaecological malignancy in developed countries (*Torre et al., 2015*). The 2016 NCCN guidelines stated that the diagnosis and treatment of endometrial cancer should be referred to imaging studies (MRI/CT/PET), although it is still surgically staged (*Koh et al., 2015*). Accurate imaging diagnosis plays an important role in the treatment and prognosis of patients with endometrial cancer (*Tsai et al., 2003*).

Corresponding author
Yuan-guang Meng,
meng6512@vip.sina.com

PET/CT and PET/MRI have provided the basis for the comprehensive evaluation of patients with endometrial cancer before surgery, and are at present an important reference for the establishment of treatment plans (*Kitajima et al., 2013*). PET/CT has been reported to play a pivotal role in restaging field (*Faria et al., 2015*; *Albano et al., 2019*), lymph node metastasis and distant metastasis of endometrial cancer (*Gee et al., 2018*). However, due to the low resolution of soft tissue in PET/CT, it is less effective than MRI at detecting the myometrial invasion of endometrial cancer (*Gallego et al., 2014*; *Kakhki et al., 2013*). This classification is important for the treatment of endometrial cancer (*Suri & Arora, 2015*). In addition, PET/CT radiation is also harmful to the patient. PET/MRI, as a multimodal molecular imaging technology, successfully integrates two imaging technologies: positron emission tomography and magnetic resonance imaging. PET/MRI, which combines the high precision of MRI and the high sensitivity of PET, has been shown to be valuable for the tumor size evaluation of endometrial cancer and lymph node metastasis (*Kitajima et al., 2009*; *Queiroz et al., 2015*; *Stecco et al., 2016*). PET/MRI currently has both integrated and sequential acquisition of PET and MRI images. However, most of the current research uses the method of post-fusion of PET and MRI images, and it is difficult to simultaneously acquire accurate PET and MRI images. The artifacts in fused image seriously degrade the imagine quality and affect the exact diagnosis.

There are few studies on the evaluation of endometrial cancer using integrated PET/MRI. The purpose of this report is to compare the evaluation efficacy of integrated PET/MRI and PET/CT on the diagnosis of endometrial cancer.

## MATERIALS & METHODS

### Patients

This retrospective study was approved by the institutional review board, and the need for patient informed consent was waived (Ethical Application Ref: GH301-18052). Our primary patient selection criteria was pathologically proven diagnoses of endometrial cancer patients who underwent pretreatment [18]F-FDG PET/CT or integrated [18]F-FDG PET/MRI for initial staging between April 2013 and May 2018. According to the primary criteria, 81 consecutive patients were selected. Of these, 37 patients underwent the [18]F-FDG PET/CT scanning, and 44 patients underwent the integrated [18]F-FDG PET/MRI scanning.

### PET/CT

Whole body PET/CT images were obtained using a PET/CT scanner Biograph 64 PET/CT (Siemens Healthcare Sector, Erlangen, Germany) as previously reported (*Stecco et al., 2016*). Patients fasted for at least 6 h before tracer injection. After injection of 2.22~4.44 MBq/kg of [18]F-FDG, PET images were obtained after an approximately 60-min uptake period with the patient's arms raised to cover the orbitomeatal line to the proximal third of the femurs. After obtaining a scout view (120-140 kVp, 30 mAs), the PET protocol comprised five to six bed positions (3 min each) depending on the patients height. Three-dimensional image reconstructions were acquired using the iterative reconstruction algorithms. The duration of PET/CT acquisition was approximately 20 min.

## Integrated PET/MRI

Simultaneous PET/MRI images were obtained using an integrated PET/MRI scanner BiographmMR (Siemens Healthcare Sector, Erlangen, Germany) as previously reported (*Schwartz et al., 2018*). Patients fasted for at least 6 h before tracer injection. After injection of 2.22~4.44 MBq/kg of $^{18}$F-FDG, simultaneous PET /MRI scan was conducted. A coronal 3D Dixon volumetric interpolated breath- hold examination (VIBE) T1-weighted imaging sequence in- and out-of -phase was acquired as a template for attenuation correction (AC). The Dixon MR acquisition, used for fat-water separation, was then segmented into four distinct tissue classifications for the MR- based AC map. The following additional MR sequences were used: half-Fourier acquire single-shot turbo spin echo (HASTE) T2 weighted imaging (axial and coronal); axial, coronal and sagittal T2; axial T2 Turbo spin echo (TSE) fat suppressed; diffusion weighted imaging (DWI) using various *b* values (50/800 s/mm$^3$); 3D axial in and out of phase; axial VIBE pre- and post- contrast and sagittal VIBE post- contrast. PET acquisition was simultaneous with MR acquisition. PET protocol comprised five to six bed positions (5 min each). No gadolinium contrast was used for the MRI portion of the study. The PET/MRI examination lasted approximately 60 min.

## Image analysis and standard of reference

Two radiologists/nuclear medicine physicians (five years of experience working in PET/CT and PET/MRI) who were especially experienced in gynecological imaging, consensually and retrospectively evaluated PET/CT and Integrated PET/MRI images for the following findings: (a) presence of the primary tumor, (b) tumor extension into the myometrium, cervical stroma, uterine serosa or adnexa, vagina or parametrium, and urinary bladder or rectum mucosa as well as (c) pelvic lymph nodes (*Kitajima et al., 2013*). Neither reader was aware of the results of other imaging studies, histopathologic findings or clinical data.

Histopathological correlation was available in all 81 patients and was used as the reference standard.

## Statistical analysis

All statistical analyses were performed using IBM SPSS version 23.0 (SPSS Inc, Armonk, NY, USA). The McNemar test was used to determine the statistical significance of differences in the accuracy of staging as determined by PET/CT and Integrated PET/MRI. Compare and analyze the data by the means of Frequencies, Crosstabs and Chi square test. Differences at $P < 0.05$ were considered to be statistically significant.

## RESULTS

PET/CT and Integrated PET/MRI examinations were successfully completed in 37 patients and 44 patients, respectively (Table 1). According to the revised International Federation of Gynecology and Obstetrics (FIGO) criteria (*Kitajima et al., 2009*), the T stage was classified as pT1a in 18 patients (PET/CT) and 24 patients (Integrated PET/MRI), pT1b in five (PET/CT) and four (Integrated PET/MRI), pT2 in five (PET/CT) and two (Integrated PET/MRI), pT3 in 2 (PET/CT) and six (Integrated PET/MRI), and pT4 in one (PET/CT)

**Table 1  Demographic and clinical characteristics of the study population.**

|  | PET/CT | Integrated PET/MRI | *p* value |
|---|---|---|---|
| **Number of patients** | 37 | 44 | / |
| **Age in years, mean (range)** | 53.2 (28–76) | 54.2 (35–73) | 0.59 |
| **BMI** | 26.2 | 24.5 | 0.23 |
| **Indication, number (%)** |  |  |  |
| Staging | 37 | 44 | / |
| Re-staging | 0 | 0 | / |
| **Treatment, number (%)** |  |  | 0.28 |
| - Surgery | 100 | 100 |  |
| Surgery only (or curettage) | 21.6 | 36.4 |  |
| With additional chemotherapy | 8.1 | 13.6 |  |
| With additional progesterone | 48.7 | 38.6 |  |
| With additional radiotherapy | 21.6 | 11.4 |  |
| - No surgery | 0 | 0 |  |
| - No treatment | 0 | 0 |  |
| - Dead before treatment | 0 | 0 |  |
| **Tumor size (cm)** | 0.6–6.5 | 0.7–7.1 | 0.84 |

**Table 2  Frequency and percentage of FIGO classification and tumor histotype of the study population.**

|  | PET/CT | Integrated PET/MRI | *p* value |
|---|---|---|---|
| **FIGO stage** |  |  | 0.29 |
| IA | 24 | 32 |  |
| IB | 5 | 4 |  |
| II | 5 | 2 |  |
| IIIA | 0 | 3 |  |
| IIIB | 0 | 0 |  |
| IIIC1 | 2 | 2 |  |
| IIIC2 | 0 | 1 |  |
| IV | 1 | 0 |  |
| **Frequency of endometrium cancer** |  |  | 0.68 |
| Adenocarcinoma G1 | 14 | 13 |  |
| Adenocarcinoma G2 | 16 | 23 |  |
| Adenocarcinoma G3 | 7 | 8 |  |

and zero (Integrated PET/MRI). The histopathologic types of the primary tumors were Adenocarcinoma: Grade 1 (14 in PET/CT, and 13 in Integrated PET/MRI), Grade 2 (16 in PET/CT, and 23 in Integrated PET/MRI), and Grade 3 (seven in PET/CT, and 8 in Integrated PET/MRI). Demographic data for the 81 patients are shown in Table 1 and Table 2.

## Primary tumor detection
Both PET/CT and Integrated PET/MRI detected 100% of the primary tumors (Table 3).

**Table 3** Parameters of diagnostic performance including FP, FN, TP, TN, sensitivity, specificity. Accuracy, PPV, and NPV of PET/CT and Integrated PET/MRI on a per-patient basis.

| | Primary tumor detection | | Regional lymph node metastasis | | Abdominal metastasis | |
|---|---|---|---|---|---|---|
| | PET/CT | Integrated PET/MRI | PET/CT | Integrated PET/MRI | PET/CT | Integrated PET/MRI |
| TP | 37 | 44 | 1 | 2 | 0 | 0 |
| TN | 0 | 0 | 31 | 40 | 36 | 44 |
| FP | 0 | 0 | 3 | 0 | 1 | 0 |
| FN | 0 | 0 | 2 | 2 | 0 | 0 |
| Sensitivity | 100 | 100 | 33.3 | 50 | 100 | 100 |
| Specificity | 100 | 100 | 91.2 | 100 | 97.3 | 100 |
| PPV | 100 | 100 | 75 | 100 | 100 | 100 |
| NPV | 100 | 100 | 93.9 | 95.2 | 100 | 100 |
| Accuracy | 100 | 100 | 86.5 | 95.5 | 97.3 | 100 |

### Regional lymph node metastasis and abdominal metastasis detection

The overall accuracy of regional lymph node metastasis detection for PET/CT and Integrated PET/MRI was 86.5% and 95.5%, respectively (Table 3). Of three positive pelvic lymph nodes, PET/CT correctly identified one as positive, with two false negative lesions, resulting in a sensitivity per lesion of 33.3%. Of four positive pelvic lymph nodes, Integrated PET/MRI correctly identified two as positive, with two false negative lesions, resulting in a sensitivity per lesion of 50.0%. Integrated PET/MRI proved more accurate than PET/CT, although the difference was not significant ( $P = 0.113$ ). Integrated PET/MRI proved significantly more sensitivity and specificity than PET/CT ( $P = 0.015$ and $P < 0.001$, respectively).

The overall accuracy of abdominal metastasis detection for PET/CT and Integrated PET/MRI was 97.3% and 100%, respectively (Table 3). When correlated with histopathology, the only one false positive of PET/CT was liver metastasis. Integrated PET/MRI proved more accurate than PET/CT, although the difference was not significant ( $P = 0.081$ ). Integrated PET/MRI proved more specificity than PET/CT.

### Deep myometrial invasion detection

PET/CT over-staged the myometrial invasion in six patients (16.2%) and under-staged it in 11 patients (29.7%). Integrated PET/MRI over-staged the myometrial invasion in three patients (6.8%) and under-staged it in5 patients (11.4%). The overall accuracy of myometrial invasion detection for PET/CT and Integrated PET/MRI was 45.9% and 81.8%, respectively (Table 4). Integrated PET/MRI proved significantly more accurate than PET/CT ( $P < 0.001$ ).

Two representative cases are shown (Fig. 1).

### DISCUSSION

This retrospective analysis compared the efficacy of PET/CT and integrated PET/MRI in the staging of endometrial cancer. PET/CT and integrated PET/MRI are similar in the diagnostic efficacy of endometrial cancer and its local lymph node metastasis. As

**Table 4 Parameters of diagnostic deep myometrial invasion of PET/CT and Integrated PET/MRI on a per-patient basis.**

|  | Deep myometrial invasion | | |
|---|---|---|---|
|  | **Overstaged** | **Understaged** | **Accuracy** |
| PET/CT | 6 | 11 | 45.9% |
| Integrated PET/MRI | 3 | 5 | 81.8% |

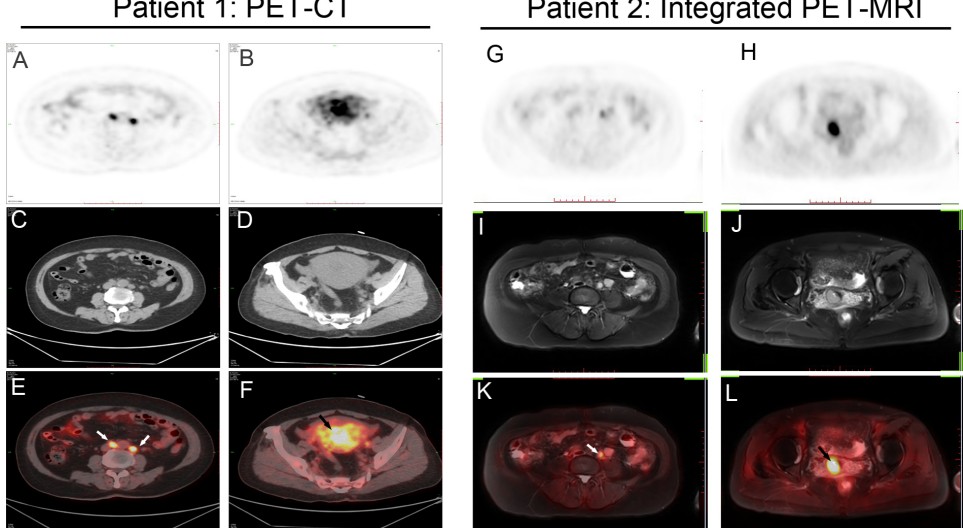

**Figure 1 Two representative cases.** (A–F) Patient 1: A 60-year-old woman with endometrial cancer and pelvic lymph node metastases. Axial PET/CT shows intense $^{18}$F-FDG uptake by uterine cavity (black arrow) and lymph nodes (white arrow), suggesting malignancy. (G–L) Patient 2: A 55-year-old woman with endometrial cancer and pelvic lymph node metastases. Axial Integrated PET/MRI shows intense $^{18}$F-FDG uptake by uterine cavity (black arrow) and lymph nodes (white arrow). Histopathologic examination confirmed cancer involvement in these lymph nodes.

expected, integrated PET/MRI is more sensitive to the diagnosis of myometrial invasion of endometrial cancer than PET/CT.

Traditional PET/CT and fused PET/MRI have been studied for tumor detection and lymphatic metastasis of endometrial cancer (*Kitajima et al., 2013*; *Queiroz et al., 2015*; *Stecco et al., 2016*). *Kitajima et al. (2013)* found that fused PET/MRI is better than MRI at diagnosing lymph node metastasis of endometrial cancer (100% vs. 66.7%). *Stecco et al. (2016)* reported that fused PET/MRI had more sensitivity, specificity and diagnostic accuracy than PET/CT for the diagnosis of lymph node metastasis. Similar to their results, our study demonstrated that integrated PET/MRI has an advantage over PET/CT in sensitivity. However, the integrated PET/MRI and PET/CT showed a consistent 100% diagnostic accuracy for tumors.

MRI is very advantageous for diagnosing localized tumors, especially high-risk factors such as tumor size, myometrial invasion, and cervical infiltration. Therefore, using the advantages of MRI, integrated PET/MRI is more advantageous than PET/CT for diagnosing

myometrial invasion of endometrial cancer (*Sawicki et al., 2018*; *Nakajo et al., 2010*). Our results provide proof. Our data are also consistent with *Duncan et al.*'s (*2012*) results, which is obtained using fused PET/MRI. Kitajima's study also found that fused PET/MRI is better than PET/CT for the diagnosis of endometrial cancer (80% vs. 60%) (*Kitajima et al., 2013*). Since integrated PET/MRI is simultaneous, its scanning time is also less than fused PET/MRI. In addition, the Integrated PET/MRI presents several challenges, including high costs of acquisition and screening, lack of standardized imaging protocols, and limitations in patients with peacemakers or claustrophobia (*Bailey et al., 2015*; *Wehrl et al., 2010*).

Increasing evidence supports the role of sentinel lymph node mapping (SLNM) for endometrial cancer (*Rossi et al., 2017*). PET/CT also appeared to improve sentinel lymph node detection in cervical and uterine cancer (*Pandit-Taskar et al., 2010*). Considering the higher accuracy of integrated PET-MRI for regional lymph node metastasis detection, we concluded that the combination of integrated PET-MRI and sentinel lymph node mapping is reasonable and feasible.

### Limitations

This study has several limitations. First, we did not allow patients to simultaneously test the PET/CT and integrated PET/MRI, but only randomly asked them to assess one method, due to patient compliance and economic reasons. This may increase the test's interference factor. Furthermore, this study is a retrospective analysis, and the number of patients according to the inclusion criteria is relatively small. Thus, our conclusions can only be considered preliminary. It is imperative to study the differences in the diagnosis of endometrial cancer between PET/CT and integrated PET/MRI using large-scale prospective clinical trials.

## CONCLUSIONS

Integrated PET/MRI imaging showed a higher application value for the diagnosis and staging of endometrial cancer diseases, but more studies are necessary to investigate its potential clinical utility.

### Funding

This work was supported by the National Natural Science Foundation of China: High-throughput sequencing identifying pathogenic genes of endometriosis (81571411). The funders had no role in study design, data collection and analysis, decision to publish, or preparation of the manuscript.

### Grant Disclosures

The following grant information was disclosed by the authors:
National Natural Science Foundation of China: High-throughput sequencing identifying pathogenic genes of endometriosis: 81571411.

### Competing Interests

The authors declare there are no competing interests.

## Author Contributions

- Li-hua Bian performed the experiments, analyzed the data, prepared figures and/or tables, authored or reviewed drafts of the paper, approved the final draft.
- Min Wang and Jing Gong performed the experiments, approved the final draft.
- Hong-hong Liu, Ming-Xia Ye and Na Wen analyzed the data, approved the final draft.
- Nan Wang analyzed the data, prepared figures and/or tables, approved the final draft.
- Wen-sheng Fan, Bai-xuan Xu and Ming-yang Wang contributed reagents/materials/-analysis tools, approved the final draft.
- Yuan-guang Meng conceived and designed the experiments, authored or reviewed drafts of the paper, approved the final draft.

## Human Ethics

The following information was supplied relating to ethical approvals (i.e., approving body and any reference numbers):

The General Hospital of PLA granted Ethical approval to carry out the study within its facilities (Ethical Application Ref: GH301-18052).

## Data Availability

The raw data are available in the Supplemental File. It includes the BMI and FIGO stage information.

## Supplemental Information

Supplemental information for this article can be found online at http://dx.doi.org/10.7717/peerj.7081#supplemental-information.

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
