# Peer review of "Comparison of integrated PET/MRI with PET/CT in evaluation of endometrial cancer: a retrospective analysis of 81 cases"

_PeerJ, doi:10.7717/peerj.7081_

## Round 0.1 · original submission · Major Revisions

Dear Authors,

The Reviewers found your manuscript very interesting, but recommend a thorough revision in order to achieve publication.

I would suggest to take into consideration the Reviewers' comments, discuss and incorporate them within your manuscript in order to reach the standard requested for publication.

Best regards

Salvatore Andrea Mastrolia
PeerJ Academic Editor

Reviewer 1 ·

Basic reporting

OK

Experimental design

OK

Validity of the findings

OK

Additional comments

Dear authors,
I think that this paper is very interesting and the topic is surely original and not well-investigated previously also due to the low number of patients studied with PET/CT.
The authors focused on the metabolic behaviour of endometrial cancer comparing 18F-FDG PET/CT and PET/MRI.
The topic presented by authors and their analysis are fascinating and with possible clinical implications both for nuclear medicine physicians and oncologist.
However, in the paper are present some points to improve and correct to let the paper more clear and complete.
After these improvements and corrections, I think that the article could be accepted.



MAIN COMMENTS

- The language should be revised by and English native speaker.
- In your work you demonstrated a detection rate of 100% for primary tumor with PET/CT and PET/MRI, a very high value. It would be interesting to have the tumor size of the primary lesion. One of the limit of PET/CT is the resolution power (about 5 mm), so it’s very important to understand the tumor size of the lesion. Are all more than 1 cm?

ABSTRACT
- You use PET/CT or PET-CT and PET/MRI or PET-MRI to explain the same thing. Please choose one of these and use it.

INTRODUCTION
- In the introduction section when you described the indications of PET/CT in endometrial cancer you did not cite the importance of PET in restaging field (which is well described in literature and probably the main indication). Please add some papers in references section; for example: Clinical and prognostic value of 18F-FDG PET/CT in recurrent endometrial carcinoma. Rev Esp Med Nucl Imagen Mol. 2018


MATERIALS AND METHODS
- You have written “Two radiologists/nuclear medicine physicians with extensive experience in gynaecological imaging” what do you mean? How many years? Are they expert on PET and CT and MRI?

RESULTS
- When you described the two groups (37 and 44), you should compare if there are differences statistically significant between the two groups considering age, stage, histological subtype,.. After these analyses you can conclude that the two groups are homogeneous for the main clinical and epidemiological features and so it is possible to compare the diagnostic accuracy of PET/CT and PET/MRI. Revise table 1 and table 2 adding a column with p value
- Please describe site of regional lymph nodes and abdominal metastases: inguinal, iliac, ...liver, ovary,...


DISCUSSION

- The advantages of PET/MRI are many and useful, but there are also some limitations of this technique: the availability, the cost, patients with peacemaker, claustrophobia,.. Please explain also the disadvantages of this method in the discussion section.



REFERENCES
- I found some references related to the topic that you did not include. Please add it. (Comparison of 18F-FDG PET/MRI and MRI alone for whole-body staging and potential impact on therapeutic management of women with suspected recurrent pelvic cancer: a follow-up study. Sawicki LM, Kirchner J, Grueneisen J, Ruhlmann V, Aktas B, Schaarschmidt BM, Forsting M, Herrmann K, Antoch G, Umutlu L.; Diagnostic performance of fluorodeoxyglucose positron emission tomography/magnetic resonance imaging fusion images of gynecological malignant tumors: comparison with positron emission tomography/computed tomography. Nakajo K, Tatsumi M, Inoue A, Isohashi K, Higuchi I, Kato H, Imaizumi M, Enomoto T, Shimosegawa E, Kimura T, Hatazawa J.).

Reviewer 2 ·

Basic reporting

1. The specific parameters of pet/ct, pet/mri and pet/mri are expected to be listed.

Experimental design

The number of samples is small. Especially the number of positive lymph nodes is too small.

Validity of the findings

We do not think that such few and controversial conclusions about positive lymph nodes should be listed

Reviewer 3 ·

Basic reporting

The text is clear, literature references are adequate. Tables must improved. English language editing is needed.

Experimental design

Research question is well defined and although the information provided are not completely novel they could be useful in clinical practice.
Methods are well described.

Validity of the findings

Although the data can not be considered conclusive, given their relevance in the clinical field, can serve as a starting point for further studies

Additional comments

The paper by Bian et al. provides a retrospective observational study with the aim to compare integrated PET/MRI with PET/CT in pre-surgical evaluation of endometrial cancer. The study population is of 81 women, a small cohort considering the high incidence rate of endometrial cancer. Moreover, many previous studies have investigated this issue, also in larger studies. Therefore, the information provided is not completely novel. However, in the era of sentinel node mapping, pre-surgical radiological staging has taken a crucial role in the management of endometrial cancer patients.
The description of the two radiological techniques appears adequate and the scientific relevance of the presented study seems adequate.
On the other hand, some criticisms have to be raised:
General points:
a) Introduction is well written, but the results section is really poor and tables may be improved with more details.
b) Please underline in the discussion section the concept that in the era of sentinel node pre-surgical evaluation has taken a crucial role in the management of endometrial cancer and that although this article cannot provide definitive conclusions, it illustrates a further useful tool to define the correct surgical planning for patients with endometrial cancer.
c) The manuscript cannot be accepted in its current form as the English language needs substantial editing.
Specific points:
a) Line 121-122: There is probably an error related to the number of patients (30 or 81?).
b) Table 1: It appears difficult to read, confusing in the organization of rows and columns.
c) Table 2 should provide information about tumor histotype. In reality, patients are only divided by grading. There is no information concerning the histotype (endometrioid, clear cells, serous, etc.). On the other hand, it’s not clear to me why the terms CIN III and Leiomiosarcoma are present, considering that there are no patients with this diagnosis and that the study is specifically focused on patients with endometrial cancer.
d) Please provide in Table 2 the division between patients with FIGO stage IIIC1 and IIIC2.

---

## Round 0.2 · accepted · Accept

Dear Authors,

I would like to compliment with you for the efforts provided in addressing the Reviewers' comments.

Your manuscript has been considered suitable for publication and can be accepted in its current form.

Best regards

Salvatore Andrea Mastrolia
PeerJ Academic Editor

Reviewer 1 ·

Basic reporting

no comment

Experimental design

no comment

Validity of the findings

no comment

Additional comments

The authors revised the manuscript following reviewers suggestions and now the paper is more clear and complete.
I suggest to accept the paper.

Reviewer 3 ·

Basic reporting

The paper can be accepted in its revised form

Experimental design

The paper can be accepted in its revised form

Validity of the findings

The paper can be accepted in its revised form

Additional comments

The paper can be accepted in its revised form